# A New Quantum Multiparty Simultaneous Identity Authentication Protocol with the Classical Third-Party

**DOI:** 10.3390/e24040483

**Published:** 2022-03-30

**Authors:** Xiang Li, Kejia Zhang, Long Zhang, Xu Zhao

**Affiliations:** 1School of Mathematical Science, Heilongjiang University, Harbin 150080, China; 2190985@s.hlju.edu.cn (X.L.); lzhang@hlju.edu.cn (L.Z.); 2190972@s.hlju.edu.cn (X.Z.); 2State Key of Laboratory of Networking and Switching Technology, Beijing University of Posts and Telecommunications, Beijing 100876, China

**Keywords:** quantum authentication, multiparty authentication, GHZ state, identity authentication

## Abstract

To guarantee information security in communication, quantum identity authentication plays a key role in politics, economy, finance, daily life and other fields. In this paper, a new quantum multiparty simultaneous identity authentication protocol with Greenberger–Home–Zeilinger (GHZ) state is presented. In this protocol, the authenticator and the certified parties are the participants with quantum ability, whereas the third party is a classical participant. Here, the third-party is honest and the other two parties may be dishonest. With the help of a classical third-party, a quantum authenticator and the multiple certified parties can implement two-way identity authentication at the same time. It reduces the quantum burden of participants and lowers down the trustworthiness, which makes the protocol be feasible in practice. Through further security analysis, the protocol can effectively prevent an illegal dishonest participant from obtaining a legitimate identity. It shows that the protocol is against impersonation attack, intercept-measure-resend attack and entangle-measure attack, etc. In all, the paper provides positive efforts for the subsequent security identity authentication in quantum network.

## 1. Introduction

In the past few years, with the rapid development of quantum computing, existing cryptographic schemes face the security threat that the scheme can not resist quantum computing attacks. In order to solve this security problem, the idea of applying quantum technology to the cryptography scheme is proposed, and then quantum cryptography appears. In quantum cryptography, the security is guaranteed by the Heisenberg uncertainty principle, quantum non-cloning theorem and other quantum mechanics principles, which is no longer based on mathematical difficulties problems. With the development of quantum cryptography, various different types of quantum cryptography protocols have been proposed, which mainly involves Quantum Key Distribution (QKD) [1,2,3], Quantum Secure Direct Communication (QSDC) [4,5,6], Quantum Authentication (QA) [7,8,9,10,11], etc. Among them, quantum authentication is becoming an important branch and has attracted more and more attention. Quantum authentication can generally be divided into the following aspects: quantum message authentication (QMA) [12,13,14], quantum entity authentication (QEA) [11,15,16], quantum identity authentication (QIA) [17,18,19,20]. Since authentication is a prerequisite for completing many quantum protocols, it will have more important application prospects in practice.

In 1999, by combining quantum key distribution and classical identification procedure, Dušek et al. first designed a secure identity authentication system [21]. In 2000, Zeng et al. put forward a quantum key verification protocol, which quantum identity authentication occurs while completing the quantum key verification [22]. In 2002, Takashi et al. presented three types of quantum identification schemes [23]. They completed two quantum identifications by using the entangled states and introducing a trusted authority. Besides, a quantum message authentication scheme was proposed via combining the quantum cryptosystem with the ordinary authentication. Until then, most QIA schemes only involved simple authentication between two or three users, but few authentications involved multiple parties. In 2006, Wang et al. proposed a multiparty simultaneous identity authentication (MSQIA) protocol based on entanglement swapping [24]. All the users in the protocol can be authenticated by a trusted third party (TTP) simultaneously. In 2013, Yang et al. proposed a quantum protocol for (t,n)-threshold identity authentication based on GHZ States [25]. In the MSQIA protocol, the trusted third party (TTP) can authenticate the users simultaneously when and only when *t* or more users among *n* apply for authentication.

In 2017, Hong et al. presented a QIA protocol based on single photons [26]. The protocol does not require any quantum memory registration and quantum entangled states to complete the authentication. In 2019, Zawadzki et al. proposed an improved version with better security for the protocol of Hong et al. [27]. The improved protocol does not require an authenticated classic channel, Bob simply confirms or denies the entire authentication transaction. In the same year, Zhang et al. presented a quantum simultaneous identity authentication based on Bell states [28]. With the help of a third party, the mutual identity authentication protocol was designed by combining Bell states and Pauli operations. This protocol can prevent a third party from knowing the originally shared key. Then, Jiang et al. proposed a mutual simultaneous identity authentication protocol between quantum user and classical user by using Bell states in 2021, which did not require the third party or complicated operations [29]. In the protocol, only the single-qubit measurement and XOR operations were performed to complete the authentication. Nevertheless, the protocols mentioned above cannot achieve multiparty simultaneous authentication.

However, in real life, it is difficult for the third-party to have quantum capability. In this paper, a new quantum multiparty simultaneous identity authentication protocol based on (r+1)-particle GHZ state the classical third-party is presented. In the protocol, the third-party does not require to prepare any quantum resources during quantum authentication communication. Moreover, the third-party only perform certain operations in the initial registration and the final certification stage, and he does not participate in the following steps. Thus, the authority of third-party is reduced and the protocol is more reasonable in reality. Furthermore, in our protocol, authenticator randomly generates quantum resource, whereas the authenticated users require to conduct measurement and reflection operations, etc.

The rest of the paper is organized as follows: in Section 2, some preliminaries are presented in this section. In Section 3, a quantum multiparty simultaneous identification protocol is proposed. In Section 4, the security analysis is described in detail. Finally, a conclusion is given in Section 5.

## 2. Preliminaries

The following basic theories needed to complete the authentication protocol. The $\left(r+1\right)$-particle GHZ state is widely used in quantum communication, it can be expressed as:(1)$${|G_{\pm }\rangle }_{12\cdots r\left(r+1\right)}=\frac{1}{\sqrt{2}}\left(|\underset{r+1}{\underset{︸}{g_{0}g_{1}\cdots g_{r}}}\rangle \pm |\underset{r+1}{\underset{︸}{\left(g_{0}⊕1\right)\left(g_{1}⊕1\right)\cdots \left(g_{r}⊕1\right)}}\rangle \right)$$
where $g_{i}\in \left\{0,1\right\}\left(i=0,1,\cdots ,r\right)$, ⊕ is the XOR operation, $|1\rangle$ and $|0\rangle$ are the two eigenstates of the Z-basis.

## 3. Quantum Multiparty Identity Authentication Protocol

In this section, we will introduce the details of our multi-party simultaneous identity authentication protocol. Alice is an authenticator, whereas $Bob_{1},Bob_{2},\cdots ,Bob_{r}$ are the certified users. We suppose that a third party, Trent, can help user Alice to simultaneously authenticate the identity of *r* legal users $Bob_{1},Bob_{2},\cdots ,Bob_{r}$. The process of identity authentication protocol is shown in Figure 1. There are no noise and losses in the quantum channel.

### 3.1. Registration

In the beginning, Alice and $Bob_{i}$(i=1,2,⋯,r) are registered with Trent, then their legal identification will be determined, respectively. In other words, each of them shares a secret identity number *K* with Trent. The secret identity number $K_{A_{0}},K_{B_{1}},K_{B_{2}},\cdots$, $K_{B_{r}}$ between Trent and Alice or $Bob_{1},Bob_{2},\cdots ,Bob_{r}$ are represented by
(2)$$\left\{\begin{array}{c} K_{A_{0}}=\left\{K_{A_{01}},K_{A_{02}},\cdots ,K_{A_{0N}}\right\}\\ K_{B_{1}}=\left\{K_{B_{11}},K_{B_{12}},\cdots ,K_{B_{1N}}\right\}\\ K_{B_{2}}=\left\{K_{B_{21}},K_{B_{22}},\cdots ,K_{B_{2N}}\right\}\\ \;\vdots \\ K_{B_{r}}=\left\{K_{B_{r1}},K_{B_{r2}},\cdots ,K_{B_{rN}}\right\} \end{array}\right.$$
where $K_{A_{0i}}\in \left\{0,1\right\}\left(i=1,2,\cdots ,N\right)$ and $K_{B_{i1}},K_{B_{i2}},\cdots ,K_{B_{iN}}\in \left\{0,1\right\}\left(i=1,2,\cdots ,r\right)$.

### 3.2. Authentication

#### 3.2.1. Preparation

Alice randomly generates a sequence of *N*
(r+1)-particle GHZ states quantum systems, each of which is in the form
(3)$$\left\{\begin{array}{c} |G_{1}\rangle =\frac{1}{\sqrt{2}}{\left(|S_{A_{01}}S_{B_{11}}\cdots S_{B_{r1}}\rangle +|\left(S_{A_{01}}⊕1\right)\left(S_{B_{11}}⊕1\right)\cdots \left(S_{B_{r1}}⊕1\right)\rangle \right)}_{A_{01}B_{11}\cdots B_{r1}}\\ |G_{2}\rangle =\frac{1}{\sqrt{2}}{\left(|S_{A_{02}}S_{B_{12}}\cdots S_{B_{r2}}\rangle +|\left(S_{A_{02}}⊕1\right)\left(S_{B_{12}}⊕1\right)\cdots \left(S_{B_{r2}}⊕1\right)\rangle \right)}_{A_{02}B_{12}\cdots B_{r2}}\\ \vdots \\ |G_{N}\rangle =\frac{1}{\sqrt{2}}{\left(|S_{A_{0N}}S_{B_{1N}}\cdots S_{B_{rN}}\rangle +|\left(S_{A_{0N}}⊕1\right)\left(S_{B_{1N}}⊕1\right)\cdots \left(S_{B_{rN}}⊕1\right)\rangle \right)}_{A_{0N}B_{1N}\cdots B_{rN}} \end{array}\right.$$
where the subscripts $A_{0m}B_{1m}B_{2m}\cdots B_{rm}$$\left(m=1,2,\cdots ,N\right)$ represent the (r+1)-particles of the *m*-th GHZ states. Alice divides all the particle of these GHZ states into (r+1) ordered sequences $S_{A_{0}},S_{B_{1}},S_{B_{2}},\cdots ,S_{B_{r}}$. Next, Alice randomly generates rN decoy photons from $\left\{|0\rangle ,|1\rangle ,|+\rangle ,|-\rangle \right\}$, and inserts *N* decoy photons into $S_{B_{1}},S_{B_{2}},\cdots ,S_{B_{r}}$, respectively. Finally, Alice holds sequence $S_{A_{0}}$ and transmits the sequences $S_{B_{1}},S_{B_{2}},\cdots ,S_{B_{r}}$ to $Bob_{1},Bob_{2},\cdots ,Bob_{r}$, respectively.

#### 3.2.2. The First Eavesdropping Detection

Once $Bob_{1},Bob_{2},\cdots ,Bob_{r}$ received sequences $S_{B_{1}},S_{B_{2}},\cdots ,S_{B_{r}}$, Alice announces the initial positions of the rN decoy qubits. Afterwards, $Bob_{1},Bob_{2},\cdots ,Bob_{r}$ store the sequence briefly. Then they select a subset of *N* decoy particles to perform the following operations: measuring the decoy photons on the Z-bases or X-bases randomly; preparing states which are same to the measured results; transmitting these decoy states from $S_{B_{1}},S_{B_{2}},\cdots ,S_{B_{r}}$ to Alice, respectively.

Once confirming that Alice has received the states, $Bob_{1},Bob_{2},\cdots ,Bob_{r}$ publish the positions, measurement results and measurement bases of the corresponding decoy photons sequence, respectively. Alice will measure these particles by using the same basis and get the measured result *R* , then compare *R* with the measured result of her initial prepared state and checks whether the results are correct.

At last, Alice computes the total error rate. If the error rate of these particles is acceptable, the protocol will continue. Otherwise, they will give up continuing to authenticate.

#### 3.2.3. Measurement and Operation

$Bob_{1},Bob_{2},\cdots ,Bob_{r}$ separately make Z-basis measurements on the $S_{B_{1}},\cdots ,S_{B_{r}}$ sequences and record the measurement results $R_{B_{1}},R_{B_{2}},\cdots ,R_{B_{r}}$. Then they perform the following operation in order according to the secret identity number $K_{A_{0}},K_{B_{1}},\cdots$, $K_{B_{r}}$, respectively. If the bit of authentication key is 0, $Bob_{1},Bob_{2},\cdots ,Bob_{r}$ will perform *X* operation on the particles which are in the sequences $S_{B_{1}},S_{B_{2}},\cdots ,S_{B_{r}}$, respectively. If the bit of authentication key is 1, $Bob_{1},Bob_{2},\cdots ,Bob_{r}$ will implement *Y* operation on the corresponding particles of sequences $S_{B_{1}},S_{B_{2}},\cdots ,S_{B_{r}}$. The specific operations and corresponding conversion results are shown in Table 1.

Next, $Bob_{1},Bob_{2},\cdots ,Bob_{r}$ insert *N* decoy photons from $\left\{|0\rangle ,|1\rangle ,|+\rangle ,|-\rangle \right\}$ into the sequence $R_{B_{1}},R_{B_{2}},\cdots ,R_{B_{r}}$, respectively. At this point, sequence $R_{B_{1}},R_{B_{2}},\cdots ,R_{B_{r}}$ is converted to sequence $R_{B_{1}}^{\prime },R_{B_{2}}^{\prime },\cdots ,R_{B_{r}}^{\prime }$. At last, $Bob_{1},Bob_{2},\cdots ,Bob_{r}$ transfer the sequences $R_{B_{1}}^{\prime },R_{B_{2}}^{\prime },\cdots ,R_{B_{r}}^{\prime }$ to Alice.

#### 3.2.4. The Second Eavesdropping Detection

Firstly, $Bob_{1},Bob_{2},\cdots ,Bob_{r}$ confirm that Alice has received the sequence $R_{B_{1}}^{\prime },R_{B_{2}}^{\prime },$ $\cdots ,R_{B_{r}}^{\prime }$. Then they announce the positions, measurement results and measurement bases of the corresponding *N* decoy photons in the sequences, respectively. After that, Alice performs the same operation as the first detection eavesdrop. Finally, Alice counts the total error rate. If the error rate exceeds the security threshold, the protocol will be terminated. Otherwise, they will continue to authenticate.

#### 3.2.5. Verification

After passing the second eavesdropping detection, sequences $R_{B_{1}}^{\prime },R_{B_{2}}^{\prime },\cdots ,R_{B_{r}}^{\prime }$ are restored to $R_{B_{1}},R_{B_{2}},\cdots ,R_{B_{r}}$ by Alice. Then she performs Z-basis measurement on the qubits at the corresponding positions of $S_{A_{0}},R_{B_{1}},R_{B_{2}},\cdots ,R_{B_{r}}$. After the measurement, according to the conversion rules are shown in Table 2, the measurement results are converted into classical results $x,{\bar{R}}_{B_{1}},{\bar{R}}_{B_{2}},\cdots ,{\bar{R}}_{B_{r}}$, which can be denoted as
(4)$$\left\{\begin{array}{c} x=\left[x_{1},x_{2},\cdots ,x_{N}\right]\\ {\bar{R}}_{B_{1}}=\left[R_{B_{11}},R_{B_{12}},\cdots ,R_{B_{1N}}\right]\\ {\bar{R}}_{B_{2}}=\left[R_{B_{21}},R_{B_{22}},\cdots ,R_{B_{2N}}\right]\\ \vdots \\ {\bar{R}}_{B_{r}}=\left[R_{B_{r1}},R_{B_{r2}},\cdots ,R_{B_{rN}}\right] \end{array}\right.$$

Then Alice publishes the results of $Q_{j}=x_{j}⊕y_{j}⊕z_{j}$, where $x_{j}$ is the measurement result of $S_{A_{0}}$, $y_{j}=R_{B_{1j}}⊕R_{B_{2j}}⊕\cdots ⊕R_{B_{rj}}$, $z_{j}=S_{A_{0j}}⊕S_{B_{1j}}⊕S_{B_{2j}}⊕\cdots ⊕S_{B_{rj}}$ (j=1,2,⋯,N and ⊕ is the XOR operation). Afterward, Trent calculates $Q_{j}^{\prime }=K_{A_{0j}}⊕K_{B_{1j}}⊕\cdots ⊕K_{B_{rj}}$. If $Q_{j}^{\prime }=Q_{j}$, Alice and $Bob_{1},Bob_{2},\cdots ,Bob_{r}$ will be seen as legitimate participants. Otherwise, there will be illegal communicators in the protocol. Finally, Trent announces to Alice and $Bob_{1},Bob_{2},\cdots ,Bob_{r}$ whether the certification is successful.

## 4. Security Analysis

Security is the most important part of quantum communication protocols. In this section, the security of the multiparty identity authentication protocol is discussed. During the transmitting procedure of quantum signals, there may be an eavesdropper who wants to pass the identity authentication by illegal operations. In general, eavesdroppers are divided into two situations, which are internal eavesdropper and external eavesdropper. Next, the security of the protocol is analyzed for both aspects.

### 4.1. Internal Attack

#### 4.1.1. Impersonation Attack

In the proposed quantum multiparty identity authentication protocol, Alice is the authenticator and resource provider, whereas $Bob_{1},\cdots ,Bob_{r}$ play the authenticated roles. In this subsection, $Bob_{e}$ is one of $Bob_{1},\cdots ,Bob_{r}$ and he may have two methods to execute the impersonation attack.

On the one hand, we suppose that attacker $Bob_{e}$ attempts to impersonate verifier Alice. $Bob_{e}$ randomly generates quantum states and allocates entangled particles to $Bob_{1},\cdots ,Bob_{r}$. In the paper, $Bob_{e}$ can follow the protocol steps faithfully, but he tries to extract the authentication keys between Trent and Alice. When $Bob_{e}$ proceeded to the Section 3.2.5, he could only perform random XOR operations on qubits due to his ignorance of the pre-shared key $K_{A_{0}}$. He also does not know the measurement results *x* of sequence $S_{A_{0}}$. If Bob wants to publish the calculation results $Q_{i}$, he will need to randomly choose one of the classical bit values of 0 or 1 to perform the XOR operations. Therefore, the probability that $Bob_{e}$ can successfully impersonate Alice is $\frac{1}{2^{N}}$. As shown on the left of Figure 2, when the number *N* of particles is large enough, the probability $P_{1}=1-\frac{1}{2^{N}}$ of failure of $Bob_{e}$ approximates 1.

On the other hand, attacker $Bob_{e}$ may impersonate the legitimate user $Bob_{j}$(e≠j). Firstly, $Bob_{j}$ has previously registered his identity information with Trent. That is, he has shared the secret identity key with Trent. In Section 3.2.3, $Bob_{e}$ requires to perform corresponding operation on the measurement result $R_{B_{j}}$ by combining $Bob_{j}$’s identity numbers $K_{B_{j}}$ and the transition rules of Table 1. Next, although $Bob_{e}$ knows the conversion rules, he is ignorant of $Bob_{j}$’s identity $K_{B_{j}}$. Hence he can perform *X* or *Y* operations on the received sequence randomly. The probability of choosing either the correct operation or the incorrect operation is $\frac{1}{2}$. Besides, the probability that the $Bob_{e}$ gets the correct conversion result is $\frac{1}{2^{N}}$. Finally, the probability $P_{2}=1-\frac{1}{2}\times \frac{1}{2^{N}}$ of $Bob_{e}$’s attack being found tends to be 1 in the Figure 2. Therefore, the protocol can effectively resist impersonation attacks.

#### 4.1.2. Entangle and Measure Attack

Moreover, we discuss whether some illegal users can get secret information through entanglement measurement attack in the process of information interaction. When the qubits are sent from Alice to $Bob_{j}$, we suppose $Bob_{E}$ performs operation $U_{E}$ on the system composed of decoy photons $\left\{|0\rangle ,|1\rangle ,|+\rangle ,|-\rangle \right\}$and the ancillary state which is prepared by $Bob_{E}$ as $U_{E}$. We can get
(5)$$\begin{array}{c} U_{E}|0\rangle |e\rangle =a|0\rangle |e_{00}\rangle +b|1\rangle |e_{01}\rangle \\ U_{E}|1\rangle |e\rangle =c|0\rangle |e_{10}\rangle +d|1\rangle |e_{11}\rangle \end{array}$$
(6)$$\begin{array}{c} U_{E}|+\rangle |e\rangle =\frac{1}{\sqrt{2}}\left[|0\rangle \left(a|e_{00}\rangle +c|e_{10}\rangle \right)+|1\rangle \left(b|e_{01}\rangle +d|e_{11}\rangle \right)\right]\\ =\frac{1}{2}\left[|+\rangle \left(a|e_{00}\rangle +b|e_{01}\rangle +c|e_{10}\rangle +d|e_{11}\rangle \right)+|-\rangle \left(a|e_{00}\rangle -b|e_{01}\rangle +c|e_{10}\rangle -d|e_{11}\rangle \right)\right] \end{array}$$
(7)$$\begin{array}{c} U_{E}|-\rangle |e\rangle =\frac{1}{\sqrt{2}}\left[|0\rangle \left(a|e_{00}\rangle -c|e_{10}\rangle \right)+|1\rangle \left(b|e_{01}\rangle -d|e_{11}\rangle \right)\right]\\ =\frac{1}{2}\left[|+\rangle \left(a|e_{00}\rangle +b|e_{01}\rangle -c|e_{10}\rangle -d|e_{11}\rangle \right)+|-\rangle \left(a|e_{00}\rangle -b|e_{01}\rangle -c|e_{10}\rangle +d|e_{11}\rangle \right)\right] \end{array}$$
where $|e_{00}\rangle ,|e_{01}\rangle ,|e_{10}\rangle ,|e_{11}\rangle$ belong to the Hilbert space of $Bob_{E}$’s probes and${\left|a\right|}^{2}+{\left|b\right|}^{2}+{\left|c\right|}^{2}+{\left|d\right|}^{2}=1$. After transmission, $Bob_{E}$ measures ancillary qubit to get $Bob_{j}$’s operations. In order to pass the eavesdropping detection without introducing any errors, he should perform the following actions:(8)$$\left\{\begin{array}{c} b=c=0\\ a|e_{00}\rangle +b|e_{01}\rangle -c|e_{10}\rangle -d|e_{11}\rangle =0\\ a|e_{00}\rangle -b|e_{01}\rangle -c|e_{10}\rangle +d|e_{11}\rangle =0 \end{array}\right.$$

However, if b=c=0, it means $a|e_{00}\rangle \;=\;d|e_{11}\rangle$. It shows that $Bob_{E}$ cannot distinguish between $a|e_{00}\rangle$ and $d|e_{11}\rangle$. Hence, the proposed protocol can resist the entangle-measure attack.

#### 4.1.3. Intercept–Measure–Resend Attack

Actually, $Bob_{e}$ as one of $Bob_{1},\cdots ,Bob_{r}$ can only get his identity number from the third party Trent. Now, we consider whether he can get the identity of $Bob_{j}$$\left(e\neq j\right)$. First of all, he could not have obtained any related information about the identity of $Bob_{j}$ by accessing Trent since the third party is absolutely honest with our protocol. Furthermore, he also can’t get the true identity of $Bob_{j}$ through the intercept–measure–resend attack.

In Section 3.2.1, Alice inserts rN decoy photons into the sequences $S_{B_{1}},S_{B_{2}},\cdots ,$$S_{B_{r}}$ each for the eavesdropping detection, respectively. However, $Bob_{e}$ does not know the initial positions and initial states of the decoy particles in the sequence Alice sent to $Bob_{j}$. $Bob_{e}$ is a quantum participant who can perform measurement operations on the Z-basis and X-basis. Therefore, if he wants to intercept the particle that Alice is transmitting to $Bob_{j}$ in Section 3.2.1, the measurement based on Z-bases and X-bases randomly can be performed. There are four measurements for these bases, |1〉, |0〉, |+〉and|−〉. Furthermore, it is difficult to just select the correct *N* position in the 2N sequence. His probability of success is $\frac{1}{2}\times \frac{1}{4^{N}}=\frac{1}{8^{N}}$.

As shown on the left of Figure 3, when *N* is large enough, the probability $P_{3}=1-\frac{1}{8^{N}}$ is approximate to 1. Therefore, it is almost impossible for illegal behavior of $Bob_{e}$ not to be detected.

### 4.2. External Attack

Unlike internal attackers, external attackers are illegal eavesdroppers from the outside. Eve is an eavesdropper who wishes to obtain some secret information to pass the identity authentication. Eve often uses the impersonation attack, the entangle and measure attack and the intercept–measure–resend attack, etc. The security of some attacks is analyzed below.

We assume that Eve tries to impersonate $Bob_{i}$(i=1,2,⋯,r). In the Section 3.2.2, Alice inserted rN decoy particles into the sequence and sent them to $Bob_{1},Bob_{2},\cdots ,Bob_{r}$ for detection eavesdropping, respectively. After Eve receives the particles, she randomly measures and sends the qubits to Alice. Moreover, Eve randomly measures particles based on Z-basis or X-basis since she dose not know the authentication key sequence $S_{B_{i}}$ shared only by Alice and $Bob_{i}$. The probability of her choosing the right operation is $\frac{1}{2}$ and the probability of picking the correct *N* particles from 2N sequence is $\frac{1}{3}$. Hence the probability of Eve being detected is $P_{4}$=$1-\frac{1}{2}\times \frac{1}{2}\times \frac{1}{4^{N}}$. As shown on the right side of Figure 3, if *N* is large enough, the probability $P_{4}$ approximates to 1. Therefore, it is difficult for Eve to pass the eavesdropping detection.

Similar to impersonation attacks, Eve is an external attacker while in the entangle and measure attack and the intercept–measure–resend attack. Eve has less information than an internal attacker, hence the probabilities of failure are higher.

## 5. Further Discussion

In this section, we compare different models to demonstrate that our protocol may be more plausible, then comparing and summarizing the quantum authentication protocols in Table 3. Through the following comparison, it can be found that most of the existing quantum authentication protocols with the third party(TP) and our proposed protocol are two different models.

**Model 1**: The model of a quantum authentication protocol with a third party is simplified as follows [11,28]: Suppose Alice and Bob are two legitimate participants who want to authenticate each other, and the TP is a third party that helps them authenticate. Before the authentication protocol begins, the legitimate Alice and the legitimate Bob share a key in advance. During the authentication process, the TP will generate the quantum states and distribute them to the participants, Alice and Bob will perform operations according to the keys. Finally, the participants confirm the identity of the other party by comparing the results published by the other party.

**Model 2**: Our proposed protocol model is simplified as follows (for the convenience of comparison, the multi-party protocol is simplified to two parties): Suppose Alice and Bob are two legitimate participants who want to authenticate each other, and the TP is a third party that helps them authenticate. Before the authentication protocol begins, Alice and Bob register their legal identities with the TP. That is, they share the secret keys with the TP, respectively. During the authentication process, Alice will generate the quantum states and distribute them to herself and Bob, and then Alice and Bob will perform certain operations based on the keys. Finally, The TP confirms the identity of the participants by comparing the calculation results of itself and Alice, and announces the results to the participants.

In practice, Model 2 is more suitable for quantum multi-party authentication than Model 1. If Alice and $Bob_{1},Bob_{2},\cdots ,Bob_{r}$ want to share keys, any two parties must share keys, which will increase a lot of unnecessary work. It is a very complicated process for each participant to share keys with each other, so we introduce a third party to actually conduct centralized key management, which simplifies the process of key distribution. Moreover, even if TP is not introduced, Model 1 can complete the mutual authentication. For example, Ref. [26] and Ref. [27] accomplish mutual authentication without introducing a third party. Therefore, in fact, our model makes more sense in practice.

Furthermore, we compare and summarize the quantum authentication protocols in Table 3. Compared with the previous quantum identity authentication, we extend the two-party authentication to multi-party authentication, which does not require all communicators to own quantum capacity. In this paper, quantum Alice and $Bob_{1},Bob_{2},\cdots ,Bob_{r}$ are able to complete identity authentication simultaneously based on (r+1)-particle GHZ states with the help of classical Trent. Only the initial registration and the final certification stage require him to perform some classical operations, and he does not participate in the rest of the time. In other words, the rights of the third party are better reduced.

## 6. Conclusions

In this paper, with the help of a classical third-party, a quantum multiparty simultaneous identity authentication protocol with GHZ state is presented. A trusted third-party centrally manages the keys of the participants, and Alice and $Bob_{1},Bob_{2},\cdots ,Bob_{r}$ complete authentication at the same time. The analysis of this protocol can effectively prevent illegal participants or attackers from obtaining legal identity information, and it can resist all kinds of ordinary attacks from the inside and outside. In addition, similar with the previous quantum multiparty simultaneous identity authentication protocol, the security analysis is based on the case of “no noise and no loss” in quantum channels [9,24,31]. In this case, our paper is also designed against the assumption of “no noise and no loss” in quantum channels. However, we need to make it clear that the security analysis under different noise rates is indeed an important content. We hope that this protocol have better application scenarios in the future.

## Figures and Tables

**Figure 1 entropy-24-00483-f001:**
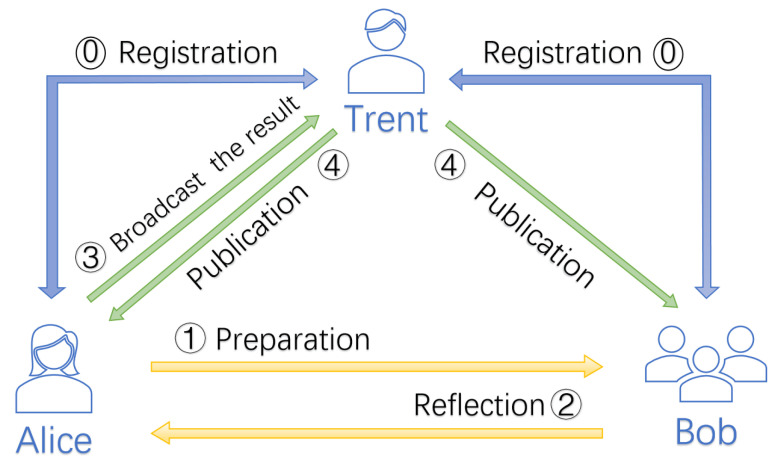
The process of quantum multiparty identity authentication protocol. (**0**): Trent shares secret keys with other users during the registration phase. (**1**): Alice sends quantum sequences to the $Bob_{1},Bob_{2},\cdots ,Bob_{r}$ separately during the preparation phase. (**2**): $Bob_{1},Bob_{2},\cdots ,Bob_{r}$ send the measured and operated particles to Alice, respectively. (**3**): At this stage, Alice announces her calculation results to Trent. (**4**): Finally, Trent compares the results to determine whether the authentication is successful and announces it to all users at the same time. At this point, the agreement is complete. In addition, the figure omits detecting eavesdropping stages for easy viewing. Nonetheless, these steps is essential in the protocol.

**Figure 2 entropy-24-00483-f002:**
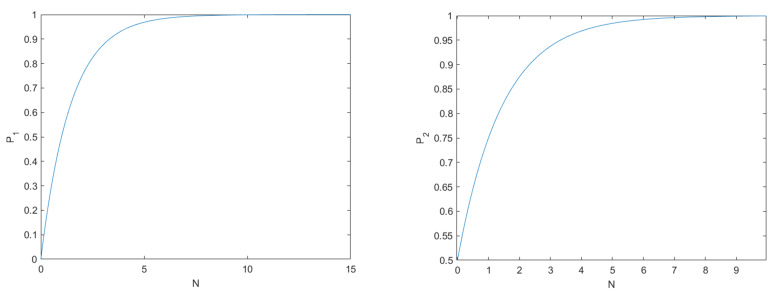
(**Left**) the probability $P_{1}$ of $Bob_{e}$ being detected. (**Right**) the probability $P_{2}$ of $Bob_{e}$ being detected.

**Figure 3 entropy-24-00483-f003:**
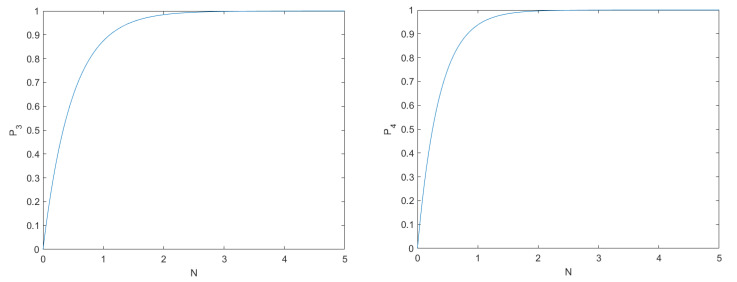
(**Left**) the probability $P_{3}$ of $Bob_{e}$ being detected. (**Right**) the probability $P_{4}$ of Eve being detected.

**Table 1 entropy-24-00483-t001:** The conversion mode of measurement result.

Quantum Bit	Opreation	Conversion Mode
bit =0	*X*: Measuring the received particles and preparing the same particles.	|0〉⟶|0〉
		|1〉⟶|1〉
bit =1	*Y*: Measuring the received particles and preparing the opposite particles.	|0〉⟶|1〉
		|1〉⟶|0〉

**Table 2 entropy-24-00483-t002:** The conversion rule of measurement result.

Measurement Result	Classical Result
|0〉	0
|1〉	1

**Table 3 entropy-24-00483-t003:** Comparison among some different quantum authentication protocols.

Protocol	Participants	The Third Party	Quantum Resource
Wang et al. [24]	Multipartite	Quantum third party	GHZ state
Yang et al. [25]	Multipartite	Quantum third party	GHZ state
Zhang et al. [28]	Mutual	Quantum third party	Bell state
Jiang et al. [29]	Mutual	No third party	Bell state
Wu et al. [30]	Multipartite	Quantum third party	Bell state and GHZ state
Our protocol	Multipartite	Classical third party	GHZ state

## Data Availability

Not applicable.

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
