# Peer review of "A New Quantum Multiparty Simultaneous Identity Authentication Protocol with the Classical Third-Party"

_entropy, 2022, doi:10.3390/e24040483_

Round 1

Reviewer 1 Report

This manuscript proposes a semi-quantum multiparty simultaneous identity authentication protocol. The proposed protocol is novel in the field to the best of my knowledge. Although the proposed protocol is correct, I can not recommend this manuscript for publishing in the journal of Entropy directly. I suggest that the authors modify and explain the following issues. 

  1. In semi-quantum protocol, the classical user only owns up to three quantum capabilities, but the classical user needs the four quantum capabilities to reach the authentication in the proposed protocol. This requirement of quantum capabilities violates the definition of a Semi-quantum environment proposed by Boyer et al.. The authors should explain why they claim that the proposed protocol is a semi-quantum protocol. 
  2. The security analysis is not rigorous enough. The manuscript only provides analyses of the well-known attacks. There is no evaluation of the amount of advantage that the eavesdropper can obtain among the different noise rates.  

Author Response

Thank you for your helpful advice. According to the article you provided, we have improved our paper. The specific modifications are summarized as follows: (1) Considering your first suggestion, we found this to be a real problem. Our protocol is actually a protocol with limited quantum ability, it is not strictly a semi-quantum protocol. Therefore,we have deleted expression of the “semi-quantum” in the revised protocol. For example, we have changed the title into " A new quantum multiparty simultaneous identity authentication protocol with the classical third-party ". We have improved and modified abstract (Line4-8), introduction (Line45-48, Line51-58, Line64-71), preliminaries (Line79-81), the process of protocol (Line86-87, Line109-118, Line124-134), security analysis (Line211-216, Line231-233) and conclusion (Line 240-245). (2) We are very sorry for your second comment. Similar with the previous quantum multiparty simultaneous identity authentication protocol, the security analysis is based on the case of “no noise and no loss” in quantum channels [9, 24, 34]. In this case, our paper is also designed against the assumption of “no noise and no loss” in quantum channels. However, we need to make it clear that the security analysis under different noise rates is indeed an important content. Its practical application is also very important, and it is indeed a problem to be studied. This is also explained in the “discussion and conclusions” section of our article, and we improved the security analysis in this paper (Line211-216, Line231-233). (3) We have corrected grammar, expression, spelling, and replaced some references (The details can be seen in page 5-7 of this paper.)

Reviewer 2 Report

The authors in the paper entitled "A new semi-quantum multiparty simultaneous identity authentication protocol" presented a new semi-quantum multiparty simultaneous identity authentication protocol with Greenberger-Home-Zeilinger (GHZ) state.
Thea uthors introduced some novelties in respect to previous semi-quantum identity authentication.
They extend the two-party authentication to multi-party authentication, which does not require all communicators to own quantum capacity. Full-quantum Alice and semi-quantum Bob1, Bob2,..., Bobr are able to complete identity authentication simultaneously basedon (r + 1)-particle GHZ states with the help of classical Trent.  Moreover, the analysis of this protocol can effectively prevent illegal participants or attackers from obtaining legal identity information.
The paper is clear, well written and can be published.

Author Response

Thank you for your helpful advice. Thank you for your review comments on our paper, for acknowledging our work, and your help is greatly appreciated. In the future, we will continue to make more efforts in quantum protocol research. I wish you good health and the best of luck.

Reviewer 3 Report

The authors proposed a new semi-quantum multiparty simultaneous identity authentication protocol. With the help of a classical third-party, full-quantum authenticator and semi-quantum participants can implement two-way identity authentication at the same time. Semi-quantum environments were proposed by Boyer et. al, in which classical Bob is restricted to perform 3 out of 4 operations only. (1.) Generating Z-basis qubits. (2.) Z-basis measurement. (3.) Reflecting photons without disturbance. (4.) Reordering photons using different delay lines. The proposed protocol applies all 4 abilities, which cannot be defined as the semi-quantum environment. Please explain the definition of the quantum environment.

Typically, semi-quantum cryptographic protocols do not involve decoy photons, due to the restriction of no quantum memory. The proposed protocol assumes Bob preserve the photon sequence while waiting for Alice to announce the position of the decoy photons. Please note storing photon sequences is beyond the ability of delay lines. Due to the issues of the protocol, unfortunately, I cannot recommend the publication of this paper in Entropy.

Author Response

Thank you for your helpful advice. According to the article you provided, we have modified our paper. The specific modifications are summarized as follows:

  • Considering your suggestion about the semi-quantum definition, we found this to be a real problem. Our protocol is actually a protocol with limited quantum ability, it is not strictly a semi-quantum protocol. Therefore,we have deleted expression of the “semi-quantum” in the revised For example, we have changed the title into " A new quantum multiparty simultaneous identity authentication protocol with the classical third-party ". Since the current article is a semi-quantum protocol, we continue to use the method of decoy state. Moreover, we have improved and modified abstract (Line4-8), introduction (Line45-48, Line51-58, Line64-71), preliminaries (Line79-81), the process of protocol (Line86-87, Line109-118, Line124-134), security analysis (Line211-216, Line231-233) and conclusion (Line 240-252).
  • We corrected grammar, expression, spelling, and replaced some references (The details can be seen in page 5-7 of this paper.)

Round 2

Reviewer 1 Report

This manuscript proposes a quantum multiparty simultaneous identity authentication protocol. The proposed protocol is novel in the field to the best of my knowledge. In addition, the authors do not persist the wrong definition that the proposed protocol belongs to semi-quantum. The authors also modified and improved the manuscript depending on our suggestions. Thus, I recommend the manuscript for publishing in the journal of Entropy. 

Author Response

(The authors gave the same response as above.)

Reviewer 3 Report

The authors proposed a new multiparty simultaneous identity authentication protocol with the classical third-party (TP). With the help of a classical TP, full-quantum authenticator and semi-quantum participants can implement two-way identity authentication at the same time.

The trustworthiness of a TP can be categorized into the four levels as follows.

  1. Trusted TP: The TP has to follow the procedure of the protocol honestly and the participants can completely trust it. Therefore, the participants can share their secret information with the TP. However, the assumption of a trustworthy TP may be impractical.
  2. Semi-honest TP: The TP has to execute the protocol loyally, but it may try to obtain the participants’ secret information passively using the records of all intermediate transmissions and computations by the participants.
  3. Almost dishonest TP: To extract the participants’ secret information, the TP may perform any possible attacks except collaborating with other participants.
  4. Dishonest TP: The TP may perform any possible attacks.

In this paper, the classical third-party is assumed to be a trusted third-party (TTP). However, identity authentication protocol has adapted to be more practical, typically does not assume the requirement of TTP. For example, the reference [33] assumes a semi-trusted third-party. In the proposed protocol, if the third-party trusted level revises into another trusted level, the protocol exists as a vital security issue. Due to the issues of the protocol, unfortunately, I cannot recommend the publication of this paper in Entropy.

Author Response

Thank you for your suggestion to our paper. According to the article you provided, our reply is as follows:

  • Considering the semi-trusted TP cited in your proposed reference [33], we briefly explain why semi-trusted TP cannot be applied in our paper.
  • We compare and analyze the difference between quantum authentication protocol using semi-trusted TP and our quantum authentication protocol. We specifically explain the significance of the existence of a third party in our protocol from two aspects: analyzing different protocol models and discussing the trustworthiness of TP. Furthermore, there is a detailed explanation of why semi-honest TP cannot be applied in our paper.
  • We illustrate the security assumptions of our protocol. These security assumptions guarantee that our protocol is secure.

Round 3

Reviewer 3 Report

The authors have satisfactorily modified their manuscript according to my previous criticisms. Therefore, I recommend the publication of this manuscript.